# Grafting and Shading—The Influence on Postharvest Tomato Quality

**Zoran S. Ilić** [1,*], **Athanasios Koukounaras** [2], **Lidija Milenković** [1], **Žarko Kevrešan** [3], **Aleksandra Bajić** [3], **Ljubomir Šunić** [1], **Renata Kovač** [3], **Elazar Fallik** [4] **and Jasna Mastilović** [3]

[1]   Faculty of Agriculture, University of Priština-Kosovska Mitrovica, 38219 Lešak, Serbia;
     lidija.milenkovic@pr.ac.rs (L.M.); ljubomir.sunic@pr.ac.rs (L.Š.)
[2]   Department of Horticulture, Aristotle University, 54124 Thessaloniki, Greece; thankou@agro.auth.gr
[3]   Institute of Food Technology, University of Novi Sad, 21000 Novi Sad, Serbia;
     zarko.kevresan@fins.uns.ac.rs (Ž.K.); alksandra.bajic@fins.uns.ac.rs (A.B.); renata.kovac@fins.uns.ac.rs (R.K.);
     jasna.mastilovic@fins.uns.ac.rs (J.M.)
[4]   Department of Postharvest Science of Fresh Produce, ARO-Volcani Center, Rishon LeZiyyon 7505101, Israel;
     efallik@volcani.agri.gov.il
*   Correspondence: zoran.ilic63@gmail.com; Tel.: +381-63-8014966

**Abstract:** Interaction of grafting and shading on tomato physical properties and chemical composition after 15 days of storage at 10 °C and 90% relative humidity was investigated in ungrafted and grafted tomato cultivars 'Optima $F_1$' and 'Big beef $F_1$' grown under shading nets (red and pearl net) and nonshaded conditions. For grafted plants 'Maxifort' rootstock was used. The effects of two weeks of storage was statistically significant when taking into account the effects of grafting, shading and variety for all tomato fruit composition parameters, except total phenols. A principal component analysis demonstrated that the changes in tomato fruit traits during the studied storage period were the main source of differentiation in tomato fruit quality. Beside a slight loss of firmness, tomato fruits were generally expected to have lower lycopene, sugar, malic and citric acid contents, higher succinic acid content, more elastic fruit skin and higher ascorbic acid content. Additionally, after storage, fruits from grafted plants had lower total phenol, higher ascorbic acid and higher succinic acid contents compared to fruits from ungrafted plants. Storage diminishes the differences in quality achieved through convenient grafting and shading combinations.

**Keywords:** *Solanum lycopersicum* L.; grafting; shading; storage; quality

## 1. Introduction

In order to meet increased consumer demand for perfect quality, excessively high temperatures and solar irradiation damage free tomatoes during the summer period. Farmers, in addition to grafting, adopt new cultivation practices such as shading, which are carried out with the aim of overcoming of production constraints [1]. Most of the grafting research is based on greenhouse production systems, and there is limited information on the compatibility with open-field cultivars and on field performance of grafted plants in various climatic conditions [2]. Low-cost protected cultivation, such as net houses, has the potential to reduce various biotic and abiotic challenges during summer production in open field, while creating a microclimate that positively affects productivity and quality of the grafted tomato [3]. Grafting can improve or reduce the fruit's external and/or internal quality, depending on the specific rootstock/scion combination. Glion et al. [4] investigated the effects of grafting in combination with various water qualities on cherry tomato yield, and on fruit quality and sensory attributes after harvest and prolonged storage.



The titratable acid concentration in tomato fruits increased in the case of grafted plants grown in the open field conditions but not for plants grown in the greenhouse [5]. The interaction between grafting and shading influences the main constituents and flavour compounds in tomato fruits [6]. Grafting did not restore the decreased concentrations of sugars and β-carotene in either scion, or volatiles in shaded tomato plants. At the same time, shading and grafting enhanced the concentration of titratable acids and certain volatiles in the tomato fruit [7].

Grafting and shading provide an alternative strategy for achieving higher fruit yield and avoiding or reducing tomato quality decrease, caused by environmental stresses, e.g., excess radiation and temperature [1,3]. However, rootstock/scion combinations affect the final size, yield and quality of fruits from grafted plants, at harvest and during prolonged storage [8].

Previously, investigators have reported that a storage temperature of 10 °C–15 °C and relative humidity 85–95% could extend the postharvest life of tomato fruits [9]. At these temperatures chilling injury and ripening rate are minimal. At the end of the storage periods (21 days at 8 °C and 90% RH), the fruit of ungrafted plants had lower weight loss, respiration rate, and phenolic acids (except for caffeic acid), but greater firmness, soluble solids content, ascorbic acid, total phenols, and antioxidant activity [10].

In one of our recent publications we analysed the details of the influence of grafting and shading on the yield, physical properties and composition of harvested tomato fruits [3]. However, information concerning the interaction of grafting and shading on tomato quality during storage is another important aspect of influence of these treatments on tomato quality traits that has not been extensively studied. Therefore, the objective of this study was to assess the impacts of grafting and shading on tomato quality traits after storage at a nonchilling temperature.

## 2. Material and Methods

### 2.1. Plant Material and Cultivation

The cultivar 'Optima $F_1$' and 'Big beef $F_1$' were grown under two colour shading nets (ChromatiNet, Nir-Yitzhak, Israel), pearl and red and without shading (control). In parallel nongrafted plants and plants grafted on 'Maxifort' (*Solanum lycopersicum* L. × *Solanum habrochaites* S. De Ruiter, Bergshenhoek, The Netherlands) rootstock were grown. Details of applied growing technology and measured conditions during the growing period are described in previously published paper [3].

### 2.2. Storage Condition

Fruit were gently hand-harvested at the mature-pink stage without calyx from third to sixth floral branch. Fruits of uniform size, firmness and colour were transported to the laboratory and randomly divided into two groups. One group was used for analysis of physical properties and composition in fresh fruits, and the other was stored for 15 days in a cooling chamber at 10 °C and 90% relative humidity. After the storage period, fruits were analysed in respect to the same quality traits as the fresh fruits. Chemical analysis was performed immediately after homogenization of samples in a blender (Food Processor 800W MCM4100GB, Bosh, Germany), and the samples were not frozen in order to avoid degradation of constituents.

### 2.3. Colour and Texture Analysis

Tomato fruit skin colour was measured with Konica Minolta Chromameter CR-400. The L* (lightness), a* (red-green) and b* (yellow-blue) values were read using D65 light source and the observer angle of 2 °C. Tomato fruit surface colour values were measured at two predetermined points at the equatorial region of twenty fruit and average values of colour coordinates were calculated. Colour difference (ΔE) between treatments was calculated according to

$$\Delta E = \sqrt{\left(L_f^* - L_s^*\right)^2 + \left(a_f^* - a_s^*\right)^2 + \left(b_{1f}^* - b_s^*\right)^2} \qquad (1)$$

where subscript *f* indicates fresh fruit, and *s* represents fruits after storage.

Texture analysis was carried for 20 randomly chosen tomato fruits out using TA.XT Plus Texture Analyser (Stable Micro Systems, England, UK). Testing conditions were the same as in details described by Milenković et al. [3].

Based on fruit firmness measurement, the share of fruits with a firmness value of under 400 N was calculated as an approximation of share of fruits that might be perceived by consumers as those that had started to soften.

### 2.4. Lycopene Content

Lycopene was extracted with acetone:ethanol:hexane (1:1:2) solution. After centrifugation at 10.000 rpm for 5 min absorbance of lycopene solution was recorded at 503 nm using a UV-Vis spectrophotometer (Cintra 303, GBC, Scientific Equipment, Dandenong, Victoria, Australia) and lycopene content was calculated using [11] formula as follows:

$$\text{Lycopene (mg/100 g FW)} = A_{503} \times 31.2/m \text{ (m-sample weight)} \tag{1}$$

where m represents sample weight

### 2.5. Ascorbic Acid

Ascorbic acid content was determined by spectrophotometric methods using 2,6-p-dichlorphenolindophenol (DIF) as a chromogenic reagent. In order to avoid degradation of ascorbic acid, immediately after the homogenization using the blender, 3 g of the sample was weighed into a centrifugation tube and 10 mL of metaphosphoric acid was added. After vigorous shaking, the sample was centrifuged at 10.000 rpm for 5 min and supernatant was used for determination of ascorbic acid. In brief, 1 mL of sample solution, 1 mL of acetate buffer and 7 mL of DIF solution were mixed and after exactly 15 s the absorbance was read at 515 nm against blank using a UV-Vis spectrophotometer (Cintra 303, GBC).

### 2.6. Total Phenol Content

Total phenol content (TPC) was determined from the extract obtained by two stage extraction procedure, with a methanol/$H_2O$ (1/1 *v/v*) in the first, and acetone/$H_2O$ (70/30 *v/v*) solution in the second stage, according to the Folin–Ciocalteu method as described by Singleton et al. [12]. In brief, aliquot of to 1 mL of combined extract was mixed with 7 mL of water and 0.5 mL of Folin–Ciocalteu reagent diluted with water (1/1; *v/v*) was added. After 5 min of equilibration, 1.5 mL $Na_2CO_3$ solution (20%; *w/v*) was added. Following 30 min, the absorbance was recorded at 730 nm using a UV-Vis spectrophotometer (Cintra 303, GBC). The results were calculated from the calibration obtained using gallic acid as the reference standard.

### 2.7. Sugars and Organic Acids

For determination of sugar and organic acid compositions high-performance liquid chromatographic (HPLC) method (Agilent 1200 series) was used. The method described by Milenković et al. (2020) [3] was used without any modifications.

### 2.8. Statistical Methods

Multivariate principal Component Analysis (PCA) was used to identify the main factors of quality variance among analysed tomato samples and to relate them to analysed fruit texture and composition parameters in order to get the insight into complex interaction of applied preharvest treatments, grafting and shading, as well as on tomato cultivar.

In order to test the significance of effects of involved treatments on tomato fruit composition main effects analysis of variance (MANOVA) was conducted for four factors examined in this experiment: grafting, shading (i.e., net colour), cultivar and storage

For all calculations STATISTICA 13 software was used (Dell Inc. (2016), Round Rock, USA. Dell Statistica (data analysis software system, version 13).

## 3. Results and Discussion

### 3.1. Physical Properties

Before analysing the effects of grafting and shading on tomato fruit composition in the postharvest period, changes in physical properties, i.e., colour and texture will be discussed, as these aspects are relevant for the consumers' acceptability of fruits.

### 3.1.1. Fruit Colour

Fruit skin colour difference between fresh (after harvest) store tomato fruits (kept under convenient conditions for 15 days) was calculated (Table 1).

**Table 1.** Colour difference ($\Delta E$) between skin colour of fresh and stored tomato fruits.

|  |  | **Control** | **Red Net** | **Pearl Net** |
|---|---|---|---|---|
| Optima | Grafted | 0.66[I] | 0.50[I] | 0.50[I] |
|  | Ungrafted | 1.65[II] | 1.64[II] | 1.77[II] |
| Big beef | Grafted | 3.35[III] | 3.91[III] | 3.17[III] |
|  | Ungrafted | 3.11[III] | 2.38[II] | 2.81[II] |

Level of difference: [III]—slightly remarkable difference; [II]—just noticeable difference; [I]—slight difference; not marked—imperceptible difference in colour.

The different ranges of $\Delta E$ [13,14] were used to compare the degree of difference. Accordingly, the values in the range of 0 to 0.5 signified an imperceptible difference in colour between two samples, 0.5 to 1.5 a slight difference, 1.5 to 3.0 a just noticeable difference, 3.0 to 6.0 a remarkable difference, 6.0 to 12.0 an extremely remarkable difference, and above 12.0 signified a colour of a different shade.

Colour changes in tomatoes harvested at the light red stage of maturity did not significantly change in the case of grafted, shaded 'Optima $F_1$' cultivar regardless of net colour, while the fruits of grafted 'Optima $F_1$' grown without shading exhibited slight differences in colour after storage under convenient conditions. In the case of ungrafted fruits of 'Optima $F_1$' cultivar, the colour changes during storage were somewhat more expressed, but the differences were subtle (Table 1). Nevertheless, the 'Big beef' cultivar seems to exhibit more pronounced changes in skin colour during storage regardless of the applied preharvest treatments. Obtained values of $\Delta E$ on the lower level are to be categorized as remarkable in the case of nonshaded fruits regardless of application of grafting, as well as in the case of shaded, grafted fruits. In the case of fruits from shaded, ungrafted plants, colour difference between fresh and stored fruits was registered to be at the upper level of just noticeable differences.

Our findings strongly suggest that colour change is cultivar dependent, and further indicate that shading may diminish colour changes in the postharvest period.

### 3.1.2. Texture Properties

Measurements of fruit firmness of fresh and stored fruits (Table 2) showed no significant changes over time. This was an expected finding since only firm fruits were stored and for a short time (15 days) under nonchilling conditions. Bearing in mind the intention to analyse influence of grafting and shading on changes in chemical composition of fruits in the supply chain, preservation of fruit firmness confirms that analyses were performed on fruits acceptable for consumers.

**Table 2.** Fruit firmness of fresh and stored tomato fruits (N).

|  |  | Control | | Red Net | | Pearl Net | |
|---|---|---|---|---|---|---|---|
|  |  | Day 0 | Day 15 | Day 0 | Day 15 | Day 0 | Day 15 |
| Optima | Grafted | 693 ± 125 | 603 ± 81 | 685 ± 105 | 612 ± 65 | 625 ± 80 | 635 ± 111 |
|  | Ungrafted | 755 ± 123 | 763 ± 174 | 716 ± 78 | 584 ± 230 | 550 ± 113 | 577 ± 214 |
| Big beef | Grafted | 723 ± 330 | 717 ± 155 | 703 ± 147 | 659 ± 33 | 730 ± 109 | 584 ± 58 |
|  | Ungrafted | 772 ± 157 | 684 ± 172 | 645 ± 30 | 534 ± 173 | 588 ± 76 | 446 ± 48 |

Although there were no significant differences in fruit firmness between fresh and stored fruits, a few other unexpected observations regarding this aspect of tomato fruit quality were made. It seems that ungrafted tomato is characterised by a more rapid decrease in firmness when grown under a red net. Similarly, 'Big beef' tomato either grafted or ungrafted is characterised by a more rapid firmness decrease when grown under pearl nets.

Furthermore, for ungrafted fruits of both cultivars grown under both shading nets the standard deviations of firmness among measured fruits were much higher in comparison to fresh fruits demonstrating that the changes of firmness of individual fruits varied more.

In order to analyse this statement more deeply, the proportion of fruits that lost firmness, % defined as the share of fruits with measured firmness value below 400 N was calculated (Table 3). Our results confirmed that ungrafted tomato fruits grown under shading nets will suffer from unequal loss of firmness resulting in heterogeneity among fruits. This change in firmness coincides with the time when it would be exposed at the market, leading to possible disapproval among consumers.

**Table 3.** Proportion of fruits that lost firmness (%).

|  |  | Control | Red Net | Pearl Net |
|---|---|---|---|---|
| Optima | Grafted | 0 | 0.00 | 0.00 |
|  | Ungrafted | 0 | 50 | 50 |
| Big beef | Grafted | 0 | 0 | 0 |
|  | Ungrafted | 0 | 30 | 65 |

Another texture parameter measured was tomato fruit skin elasticity (Table 4). Obtained results show that changes in skin elasticity are not statistically significant. However, tomato fruits grown under shading nets become more elastic during storage, indicating it can be pressed more intensively and more deeply before puncture. This result indicates that after storage, tomato fruits grown under shade nets might be perceived as somewhat softer by consumers, which may lead consumers to conclude that the fruits are riper or beginning to lose firmness.

**Table 4.** Skin elasticity (mm) of fresh and stored tomato fruits.

|  |  | Control | | Red Net | | Pearl Net | |
|---|---|---|---|---|---|---|---|
|  |  | Day 0 | Day 15 | Day 0 | Day 15 | Day 0 | Day 15 |
| Optima | Grafted | 3.3 ± 0.8 | 3.2± 0.8 | 3.0 ± 0.5 | 4.0 ± 0.6 | 3.0 ± 0.5 | 3.5 ± 0.6 |
|  | Ungrafted | 3.8 ± 0.5 | 3.5 ± 0.4 | 3.0 ± 0.6 | 3.8 ± 1.0 | 2.7 ± 0.6 | 4.0 ± 0.8 |
| Big beef | Grafted | 4.0 ± 0.6 | 4.1 ± 0.9 | 3.9 ± 1.0 | 4.4 ± 1.0 | 3.0 ± 0.7 | 4.2 ± 0.8 |
|  | Ungrafted | 3.2 ± 0.8 | 3.1 ± 0.9 | 4.0 ± 0.6 | 4.7 ± 0.7 | 3.8 ± 0.8 | 4.8 ± 0.8 |

*3.2. Fruit Composition*

3.2.1. Lycopene Content

This study confirms that tomato fruits contain significant amounts of lycopene, which may vary in the postharvest period. Lycopene is usually synthesized from phytoene and regarded as the common

precursor for β-carotene. Lycopene formation and degradation is regulated by several biosynthetic enzymes. Higher temperatures may promote activity of the enzymes involved in lycopene formation and thus result in an increase in lycopene content in stored fruits. This statement was confirmed by Toor and Savage [15] who reported that lycopene level in light-red tomatoes increased up to 3-fold at storage temperatures of 15 °C–25 °C. In contrast, storing tomato fruits at nonoptimal temperatures for enzymatic activity may reduce activities of enzymes involved in lycopene degradation and maintain lycopene contents relatively stable. Farneti et al. [16] concluded that storage of tomatoes at temperatures below 12 °C induces lycopene degradation. Toor and Savage [15] reported that storage at lower temperatures (7 °C) inhibited accumulation of lycopene in tomatoes.

The results obtained in this study, for tomatoes stored for 15 days at 10 °C, corroborate these findings in the case of cv 'Optima $F_1$' and in ungrafted cv. 'Big beef $F_1$' for which lycopene content decreased significantly during the storage period (Figure 1). For these treatments, it seems that the decrease in lycopene is somewhat more expressed in the case of tomato grown under both nets in comparison to the control fruits grown without shading. As reported by Milenković et al. [3] application of colour shade nets to tomato plants is effective in substantially improving lycopene content under excessive solar radiation during summer period, but based on the results obtained in this study, lycopene content in tomato fruits grown under shade nets may decrease more rapidly when fruits are stored at lower temperatures.

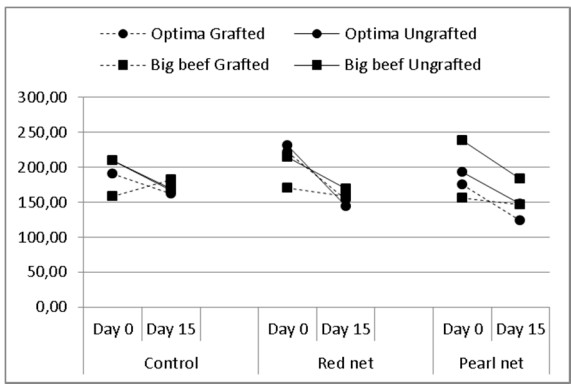

**Figure 1.** Lycopene content (g·100g$^{-1}$) in fresh and stored tomato in relation to cultivar, grafting and shading conditions.

The trend is different for the grafted 'Big beef' tomato. For the fruits grown under shading nets, a decrease in lycopene content is much less expressed, while in fruits grown without shading, this bioactive compound tends to increase (Figure 1). This observation is stressing the specific interaction between the rootstock and the scion, since the lycopene content is lower in comparison to all other treatments, regardless of shading. Lycopene content can be influenced by grafting, but it is subject to significant rootstock–scion interaction, which indicates that graft combination plays an important role. Thus, the lycopene content in grafted cv. 'Big beef $F_1$' was significantly lower (158.6 mg/100g FW) than in fruits from nongrafted plants, and the lycopene levels further decreased with shading condition. As reported earlier by Milenković et al. [3], the combination of shading and grafting produces tomato fruits with significantly lower lycopene content in both cultivars (Figure 1), and based on this study, it is clear that grafting may influence the lycopene content changes in the postharvest period too.

According to several studies, the lycopene concentration in tomato fruits tends to decrease with grafting. Helyes et al. [17] observed lower lycopene content in grafted plants, which was explained by the significantly higher yield reached by grafting. Contradictory results reported by Turhan et al. [18] state that the lycopene content does not differ significantly between the grafted and nongrafted plants. Hossain et al. [19] showed that the maximum lycopene content (0.076 mg/100 g FW) was found in

the fruits of grafted plants, whereas the minimum (0.057 mg/100 g FW) was found in the fruits of the nongrafted plants.

The available literature does not provide a clear pattern for the lycopene rates in the case of grafting. However, Nicoletto et al. [20] observed different behaviours depending on the rootstock/scion combination with landraces. Vinkovic-Vreck et al. [21] concluded that this component was not affected by grafting but mainly by the tomato variety, whereas Helyes et al. [17] obtained lower lycopene content in grafted plants, which was explained by the significantly higher yield, reached by grafting.

### 3.2.2. Ascorbic Acid and Total Phenols Content

Ascorbic acid and phenolic components are two classes of antioxidant compounds commonly found in fruits. The plant accumulates antioxidants in order to protect itself from the stressful conditions. On the other hand, their secondary metabolites boost antioxidant properties of food, increasing thus the value of fruits in human nutrition. Since shading is designed to protect plants from external stress, these parameters were investigated.

Tomato fruit contains significant amounts of ascorbic acid (AA), and the AA levels strongly depend on cultivar origin. Although light is not essential for the synthesis of AA in plants, the amount and intensity of light during the growing season have a definite influence on the final content of AA. AA is synthesized from sugars supplied through photosynthesis in plants.

There are numerous conflicting reports on changes of tomato fruit quality, including AA content, due to grafting [22]. Rahmatian et al. [23] reported that AA content increased in fruits harvested from grafted tomato plants, while several other studies have shown that AA content is strongly reduced by grafting in both, greenhouse and field studies [24,25]. Lower AA content is explained by the higher plant/shoot biomass in grafted plants compared with nongrafted ones, or by the fact that grafted plants were initially subjected to stress following the grafting operation. The decreased AA content in fruits from grafted plants could therefore be the result of redistribution or accumulation of AA in other parts of the grafted plants [26]. In our study AA levels (11.3–22.5 mg/100g FW) were lower by 21.2–29.3% in fruits from grafted plants when compared with fruits from nongrafted plants (Figure 2), and the extent of the decrease in the concentration of this bioactive component was cultivar dependent. This finding supports results reported by Qaryouti et al. [27], who found that AA content was reduced in the case of soil cultivation of cultivar Cecilia grafted on He-Man and Spirit. It is obvious that the effects of grafting onto various rootstocks on AA content may be either positive or negative, and it depends strongly on rootstock–scion compatibility.

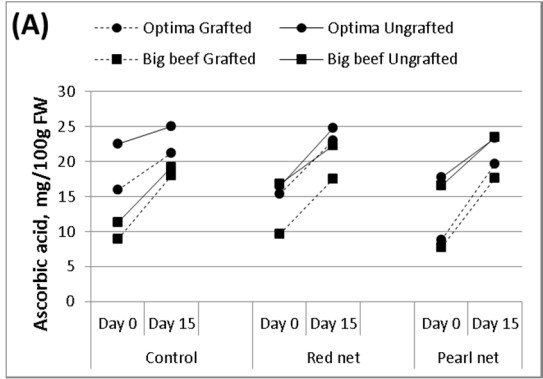 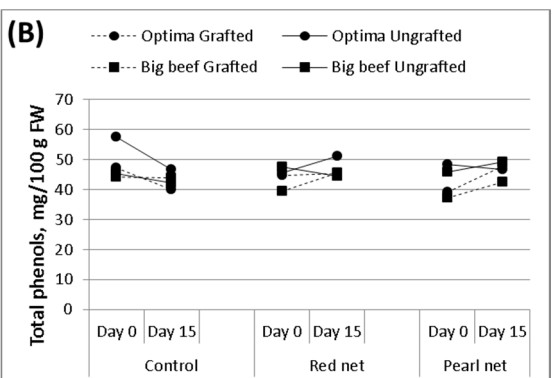

**Figure 2.** Ascorbic acid (**A**) and total phenols (**B**) content in fresh and stored tomato in relation to cultivar, grafting and shading conditions.

Among the preharvest factors influencing the final AA content in fruits, light intensity and temperature are considered as the most important. Excessive temperatures and sun irradiation cause

the triggering of defence mechanisms in plants and thus more protective component production, including AA. Thus, unshaded plants have higher AA content in comparison to the shaded ones.

This phenomenon was observed in our experiment (Figure 2). The AA content in both cultivars decreased under shade nets, with the exemption of the nongrafted cv. 'Big beef $F_1$'. AA content was specifically influenced by different colour nets, thus both cultivars under pearl nets had lower contents of AA, suggesting more pronounced protective properties of this net when compared with the red net.

The AA content of the tomato fruit increased during storage (Figure 2). The increase trend was similar regardles of growing conditions and cultivar. These results suggest that the AA synthesis in tomato continues during postharvest maturation. Accordingly, the AA content significantly increases during storage in grafted and nongrafted plants in both cultivars. Selection of the cultivars with the highest AA content is a much more important factor than climatic conditions and cultural practices in producing fruits with high amounts of AA at harvest.

In the summer season, unshaded tomatoes are exposed to excessive stress conditions, thus protective plant chemicals such as phenols accumulate more in tomato fruits grown under those conditions in comparison to shaded fruits [3]. In our experiment, total phenol (TP) content was lower in grafted plants under shading nets for both cultivars as well as for ungrafted cv. 'Optima $F_1$', while in the case of ungrafted cv. 'Big beef $F_1$', no differences between the shaded and nonshaded fruits were observed (Figure 2). The same exemption was recorded in the case of AA content, pointing out the potential of the combination of 'Big Beef' as the scion and 'Maxifort' as a rootstock to be less sensitive to stress caused by high temperature and sun irradiation.

After ten days of storage, the TP content in shaded fruits was constant compared to the levels at harvest. In tomatoes grown under shade nets that were not exposed to stressful conditions of high temperature and solar irradiarion, the initial TP content was lower and did not change significantly during tomato storage regardless of preharvest treatment. In contrast, in nonshaded control fruits, the TP content decreases during the storage period as the stress conditions are not present any more.

Vinkovic-Vrcek and co-workers [21] reported that grafting reduces both the TP content and antioxidant activities of tomato fruits. Under the standard growth conditions applied in this experiment, grafting significantly reduced the TP level only in the cv. 'Optima $F_1$'.

However, the TP content changed significantly among the scions–cultivars used in our experiment, as well as between cultivars under different shade nets.

### 3.2.3. Sugar Content

The amount and types of sugars stored in tomato fruits are among the major properties determining the postharvest quality of tomato. It affects the taste and overall fruit quality. However, the preharvest treatment applied in this experiment (grafting and shading), as well as the maturity at harvest, influence the initial sugar content in harvested tomato fruits at the beginning of storage period.

Based on our results, it was reported by several groups that sugar content was lower in tomato fruits from grafted plants in comparison to ungrafted ones. According to the results of Pogonyi et al. [28], the sugar content in tomato fruits cv. 'Lemance $F_1$' was up to 25% lower for tomato plants grown in soil culture and grafted on 'Beaufort' rootstock compared to ungrafted plants. Based on our previously published results Milenković et al. [3] we have shown that grafting is the main factor determining differences in both fructose and glucose content and their sum in the case of cv. 'Optima $F_1$' and 'Big beef $F_1$' grafted on 'Maxifort'. We showed that the sugar content in fruits from grafted plants was significantly lower ($p < 0.05$) in comparison to ungrafted plants for tomatoes grown under both red and pearls shading net. In the case of unshaded tomatoes, the sugar content was also lower in grafted plants, however this difference was not statistically significant.

The reason for lower sugar content in grafted tomatoes could be related to the rootstock effect on scion vigour, timing of flowering, fruit load, yield and, ultimately, fruit maturation, as fruit sugar concentration is highly dependent on fruit maturity at harvest [29]. Changes in dry matter content could, beside photosynthesis, also be one of the reasons for the reduced sugar concentration in grafted

plants [6]. Namely, water uptake-efficient rootstocks may increase fruit water content leading to a reduced fruit sugar concentration.

Similarly, our experiment revealed that the total sugar content was initially higher in ungrafted tomatoes (Figure 3A). During storage, regardless of grafting and shading, the total sugar content decreased with time, and more markedly in ungrafted tomatoes grown under the pearl net. A more rapid sugar decrease trend was also noted in the ungrafted tomato cultivar 'Optima' grown under red nets (Figure 3A). A decrease in total sugar content may contribute to a dull taste of tomato fruits after storage. However, for all treatments, the glucose/fructose ratio slightly decreased during the storage period (Figure 3B), revealing a more striking decrease in glucose content, while the fructose content stayed constant. It is noteworthy that the sweetness index for fructose is higher (140) than for glucose (70), and while the total sugar decrease is modest, the loss of fruit sweetness is more pronounced.

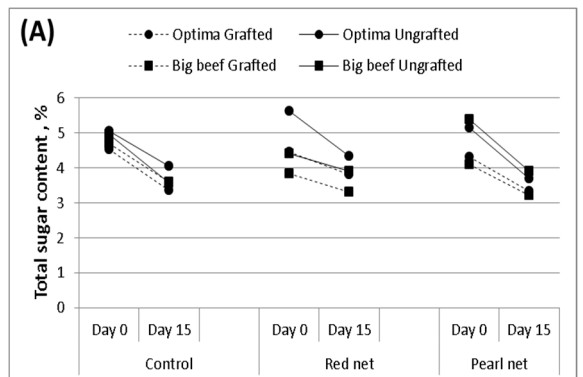 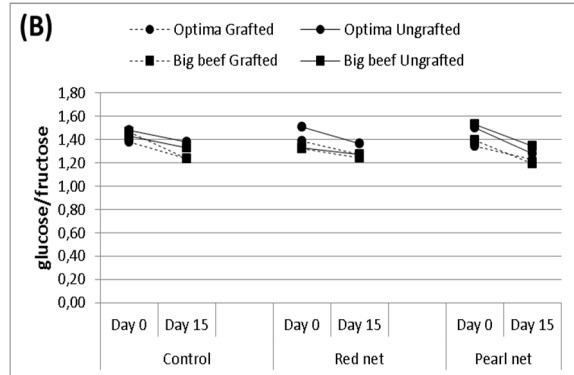

**Figure 3.** Sugar content (**A**) and glucose/fructose ratio (**B**) in fresh and stored tomato in relation to cultivar, grafting and shading conditions.

### 3.2.4. Acid Content

The organic acid content in tomato fruit ranges from as low as 0.2 to over 1.5 g·100g$^{-1}$ fresh fruit weight. In composition of acids in tomato fruit, three major acids are represented: citric, malic and succinic. Tomato sourness is an important component of tomato taste and depends not only on total acid level but also on the composition of acids, due to the differences in sensory sensations during tomato consumption that these acids provoke. The taste of citric acid is described as tart delivering a "burst" of tartness; malic acid is characterized with smooth tartness, while succinic acid, beside tartness, also provides a note of bitterness [30].

Krumbein and Schwarz [6] reported that titratable acidity was influenced by both shading and grafting, exhibiting an increase by 9 and 6%, respectively. In our previously published work [1,3]) we showed that application of shading nets significantly influences total acid content as well as citric and malic acid contents individually ($p < 0.001$), while the effects of the pearls net causes an increase in malic acid. Moreover, we reported that the only acid content and ratio significantly influenced by grafting was citric acid. Other authors also reported that oppositely to its influence on sugar content, grafting did not influence the acidity of fruits [28,31].

When comparing total acid content (Figure 4A) and the contents of citric (Figure 4B), malic (Figure 4C) and succinic (Figure 4D) acids, it is clear that 15 days of storage resulted in changes in both acid content and composition. At the end of postharvest period, total acid content (Figure 4A) decreased in tomatoes grown under the pearls shading net, while in the case of the red net, it did not change during storage. In the case of nonshaded fruits the trend was cultivar dependent with decrease in cv. 'Big beef F$_1$', and it slightly increased in cv. 'Optima F$_1$'. Trends in total acid content are the outcome of different trends in individual organic acid content. Citric acid content (Figure 4B) decreased rapidly during storage in nonshaded fruits and fruits grown under the pearl net, while in of the fruits grown under the red net, it did not change. Based on these results, it can be assumed that

the "bursting" tartness originating from citric acid, characteristic for incompletely ripe tomato fruits, might be reduced in tomato fruits after storage which was not shaded or which was shaded under the pearl net. Contrarily, in tomatoes grown under the red net, this sensation might be preserved after two weeks of storage. Malic acid content (Figure 4C) slightly decreases in the case of all examined treatments, while succinic acid content (Figure 4D) increases sharply. It can be assumed that the increased succinic and decreased malic acid content might contribute to less smooth tartness of tomato fruits, giving them a possible sensation of bitterness. However, these assumptions have to be confirmed or rejected through sensory testing in future experiments.

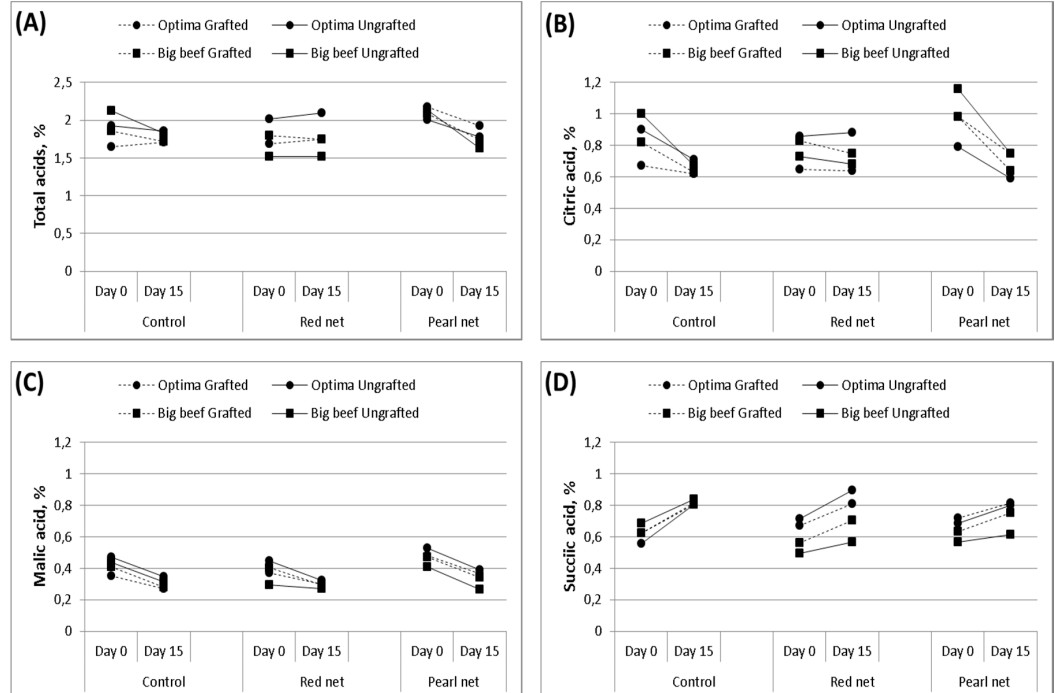

**Figure 4.** Total acid content (**A**) with citric (**B**), malic (**C**) and succinic (**D**) acid content in fresh and stored tomato in relation to cultivar, grafting and shading conditions.

### 3.3. Significance of Effects

In order to test the significance of effects described above, main effects ANOVA was conducted for four factors examined in this experiment: grafting, shading (i.e., net colour), cultivar and storage (Table 5). Our results suggested that the influence of grafting was significant for all examined tomato fruit composition parameters, except for succinic acid. Shading exhibited significant influence only in the case of total sugars content, while cultivar significantly influenced ascorbic acid content, total acid content, malic and succinic acid contents. Exposure of tomato to two weeks of storage significantly influenced all tomato fruit composition parameters, except the total phenols.

**Table 5.** Significance of influence (p values) of investigated factors on chemical composition of tomato fruits (MANOVA—main effects analysis of variance).

|  | Lycopene, mg/100g FW | Ascorbic Acid mg/100g FW | Total Phenols mg/100g FW | Total Acids, % | Citric Acid, % | Malic Acid, % | Succinic Acid, % | Total Sugars, % |
|---|---|---|---|---|---|---|---|---|
| Grafting | *** | *** | *** | * | *** | ** | ns | * |
| Shading | ns | ns | ns | ns | ns | ns | ns | * |
| Cultivar | ns | *** | ns | * | ns | *** | *** | ns |
| Storage | *** | *** | ns | * | *** | *** | *** | *** |

*** significant for $p < 0.001$; ** significant for $p < 0.01$; * significant for $p < 0.05$; ns—not significant.

### 3.4. Principal Component Analysis

Clearly the changes in physical properties and composition of tomato fruits during storage depend on a complex interaction of applied preharvest treatments, grafting and shading, as well as tomato cultivar. In order to further delineate these complex relations, we performed a principal component analysis (PCA) (Figure 5). Parameters with similar trends (glucose and fructose having same trend like total sugars) and minor influence (total acids) were omitted in the analysis. The results are presented in a factorial plane as a bi-plot (Figure 5). The first two principal components explain 60.95% of total variability (PC1: 41.04% and PC2: 19.91%).

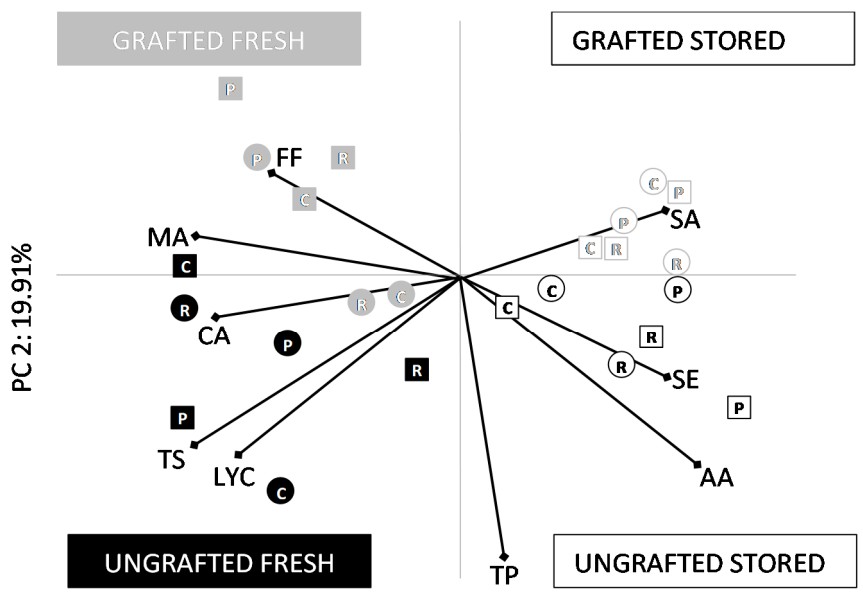

**Figure 5.** Principal component analysis (PCA) analysis of composite effects of grafting and shading on tomato fruit composition at the moment of harvest and after the storage period. Legend: Cultivars: square Big beef, circle Optima, Grafting: black ungrafted, grey grafted, Shading: C—control, nonshaded; R—red shading net; P—pearl shading net, Storage: solid marker fill—fresh fruits; white marker fill—stored fruits, Quality traits: LYC—lycopene; AA—ascorbic acid; TP—total phenols; TS—total sugars; CA—citric acid; MA—malic acid; SA—succinic acid; FF—fruit firmness; SE—fruit skin elasticity.

The distribution of treatments in the factorial plane, with all fresh tomato samples positioned in the left half of factorial plane and the stored tomato samples on the right side, demonstrate that the changes in tomato fruit traits during the storage period are the main sources of differentiation of tomato fruit quality. The main traits differentiating fresh tomato fruits, as observed from the bi-plot distribution are firmer fruit, higher sugar content, higher malic and citric acid content, and higher lycopene content. Fruits after two weeks of storage are characterized by higher succinic acid content, more elastic fruit skin, and higher ascorbic acid content.

The differences between grafted and nongrafted fruits are also visible, but minor in comparison to the influence of the storage process. All samples of tomato fruit from grafted plants are positioned in the upper part of the factorial plane, whereas samples of fruits from ungrafted plants are positioned in the lower factorial plane part. The position of the tomato fruit traits reveal that fresh grafted tomato fruits can be related to higher fruit firmness, while in the grafted fruits after storage, higher succinic acid content might be expected. Ungrafted fruits may have higher total phenol content in comparison to grafted ones; in fresh fruits, ungrafted higher sugar and lycopene content might be expected; while in ungrafted fruits after storage, ascorbic acid content might be higher.

Another noteworthy finding from the PCA plot is that tomato fruits after storage are more densely positioned in comparison to fresh fruits. We therefore conclude that storage can diminish the

differences in quality achieved through convenient grafting and shading combinations. However, one aspect of differentiation can be noted. Unshaded tomatoes are positioned further away from tomatoes grown under shading nets, indicating that in unshaded tomatoes, the changes characterizing stored fruits occur to a lesser extent. The increase in succinic acid content and skin elasticity, but also the ascorbic acid content in nonshaded fruits might be expected to be lower in comparison to shaded fruits after storage. After a storage period of 15 days, the differences that could be attributed to the variety and shading conditions were less expressed in comparison to their expression in fresh fruits. Samples of fresh fruits are distributed on a much larger area and farther away from each other, indicating that higher variability in respect to diverse quality traits due to applied agrotechnical measures can be expected in fresh tomatoes, but the variability will be lost during prolonged storage.

## 4. Conclusions

The present study investigated the relationship between shading and grafting and changes in various physical properties and chemical compositions of tomato fruits that were stored for a period of two weeks at a low, nonchilling temperature. Several conclusions were made, that provide directions for optimal utilization of these two growing techniques in the case of the tomato, which is exposed to prolonged storage in the supply chain.

Since the fruits were harvested at the mature pink ripening stage, only slight changes in colour properties were registered. Differences in fruit firmness between fresh and stored fruits were not statistically significant, but it seems that ungrafted tomatoes grown under shading nets may be prone to a more rapid decrease in firmness. Furthermore, the observed loss of firmness varies with individual fruits, resulting in heterogeneity among tomato fruits with regards to firmness after storage. In the case of fruits grown under shading nets, the skin becomes more elastic during storage in comparison to fruits from nonshaded plants.

Our principal component analysis indicated that after storage, fruits will be characterized by lower lycopene, sugar, and malic and citric acid contents. The decrease in lycopene content is more expressed in tomatoes grown under both shading nets.

Total sugar content, regardless of applied grafting and shading treatments, decreased uniformly. Taking this observation into account and the finding that total sugar content was higher in ungrafted tomatoes prior to storage, these tomatoes were also characterised with preserved higher sugar contents after storage. Based on our results showing that for all treatments the glucose/fructose ratio slightly decreased during storage thus resulting in a higher ratio of fructose/sucrose, a loss of fruit sweetness might be expected to be less expressed than the total sugar decrease.

The storage period resulted in changes in both acid content and composition. Citric acid content decreased rapidly during storage in the cases of nonshaded fruits and fruits grown under the pearl net, while the red net did not induce any change, indicating that the "bursting" tartness originating from citric acid might disappear after storage of tomatoes produced without shading or under the pearl net, while in the case of tomatoes grown under the red net, this sensation would probably be preserved even after two weeks of storage. Malic acid content decreased slightly in the case of all treatments, while succinic acid content increased sharply. The decrease in malic acid content might contribute to a less smooth tartness of tomato fruits and a possible sensation of bitterness due to the increase in succinic acid content. Although it is not confirmed, based on individual results, our principal component analysis indicated that higher succinic acid content might be expected after storage of fruits from grafted plants in comparison to fruits from ungrafted plants. This points to a higher probability for the occurrence of a bitter taste originating from the succinic acid after storage of tomatoes from grafted plants.

The ascorbic acid content of the tomato increases during storage regardless of growing conditions and cultivar, while the total phenols content remains at about the same level when compared to the level at the harvest time.

Based on the results of main effects ANOVA, exposure of tomatoes to two weeks of storage significantly influenced all tomato fruit composition parameters, except total phenols. Principal component analysis pointed out that the changes in tomato fruit traits during the storage period are the main source of differentiation of tomato fruit quality, and that storage diminishes the differences in quality achieved through convenient grafting and shading combinations.

The results presented here provide the first indicative directions for tomato growers and distributors regarding applications of shading nets and the grafting of tomatoes as growing techniques to increase yields and to reduce sun damage in fruits during the summer period. Cultivating tomatoes under shading nets may result in uneven firmness after prolonged storage, especially in the case of pearl nets. The lycopene content decrease in mature pink harvested fruits is more expressed in fruits from plants grown under shading nets. Grafted tomatoes are characterized by lower sugar content, both after harvest and after storage. The increase in succinic acid during storage, resulting in possible bitterness, may similarly be more expressed in fruits from grafted plants.

Changes in tomato fruits during storage related to sugars and acid content and composition expose probable differences in sensory properties of tomato fruits after storage and are correlated with shading and grafting, which must be further analysed and confirmed in future investigations.

**Author Contributions:** Z.S.I., J.M., A.K., and E.F. Head of the research group planned the research, analysed, and wrote the manuscript; L.M. and L.Š. conducted the experiment in the field; Ž.K. conducted the postharvest experiment; and R.K. and A.B. performed analyses of physical properties and chemical composition in the laboratory. All authors have read and agree to the published version of the manuscript.

**Funding:** This study: which was part of the projects TR-31027, TR-34012 and III 46001 was financially supported by the Ministry of Education Science and Technological Development of the Republic of Serbia.

**Conflicts of Interest:** The authors declare no conflict of interest.

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
