# Peer review of "Grafting and Shading—The Influence on Postharvest Tomato Quality"

_agriculture, doi:10.3390/agriculture10050181_

Round 1
Reviewer 1 Report
MS 'Grafting and shading - the influence on postharvest tomato quality' describe very interesting study regarding the interaction if grafting and shading on changes of tomato physical and chemical traits during 15 days of storage. Some significant effects were observed regarding all fruit tomato composition parameters except phenols. Some of the components were analyzed using UV-VIS Spectrophotometer (lycopene, ascorbic acid, total phenols), while sugars and organic acid were detected using HPLC. Using the PCA they pointed out that that the changes in tomato fruit traits during storage period are the main source of differentiation of tomato fruit quality ant that the storage diminishes the differences in quality achieved through convenient grafting and shading combinations.
The results of this study are very interesting for farmers as well as for readers of this journal as data provide new knowledge about the effects of shading and grafting on tomato fruit characteristics changes during storage. Authors also pointed out some undesirable changes of tomato fruit traits that can occur during storage and were probably a consequences of observed treatments.
Author Response
Dear review 1
Please open Attach and recognised our answer!!

Reviewer 2 Report
Manuscript is well written but has some spelling errors that should be checked.
Also, M&M section should clarify some used methods and describe it like it is usually done for that type of measurements as shown way can lead to comparison with different methods in literature and show huge difference in results measurement.
Regards,

Author Response
Dear review 2
Please open Atach and recognised our answers !!!!

Reviewer 3 Report
Dear Editor!
I checked the mamuscript 'Grafting and Shading—the Influence on Postharvest Tomato Quality'. Manuscript presents an interesting results regarding the application of net shades and grafting and their mutual effects on physico-chemical parameters of two tomato cultivars. Introductory part is focused in the paper, methodology is given in detail, while results and discussion needs some improvements. Authors should consider comments below, among others, when they are improving the manuscript.
English language is acceptable but there are some ackward sentences that must me corrected:
lines 207-209
lines 362-363
errors:
line 315 irradiation
line 317 oppositely
Abstract
I suggest taht results regarding color changes might be added in abstract
line 22: storage had lower
Introduction
Lines 60-63 I suggest a more detailed aims of study
Materials and Methods
I suggest to state that experiment was an open field carried out in net tunnels in area ....
line 103: g instead of rpm
line 109: specify the concentration of metaphosphoric acid
Results and Discussion
line 203 Table 3: specify in table title below 400 N
line 211 Table 4: specify in table title units of elacticity (mm)
line 248 capture of Fig. 1. specify the unit for lycopene contents
line 334 Based on our
line 340: to be statistically
line 377 acid content and their ratio
Principal component analyse: I suggest to deduce the most influential parameters according to PCA that allow clustering of traits (if possible)
Conclusion
According to my opinion concluision is too long, I suggest to shorten it.
Author Response
Dear review 3
Open Attach and recognised our answers!!!!1
